# ReLAM: Learning Anticipation Model for Rewarding Visual Robotic Manipulation

## Abstract

Reward design remains a critical bottleneck in visual reinforcement learning (RL) for robotic manipulation. In simulated environments, rewards are conventionally designed based on the distance to a target position. However, such precise positional information is often unavailable in real-world visual settings due to sensory and perceptual limitations. In this study, we propose a method that implicitly infers spatial distances through keypoints extracted from images. Building on this, we introduce Reward Learning with Anticipation Model (ReLAM), a novel framework that automatically generates dense, structured rewards from action-free video demonstrations. ReLAM first learns an anticipation model that serves as a planner and proposes intermediate keypoint-based subgoals on the optimal path to the final goal, creating a structured learning curriculum directly aligned with the task's geometric objectives. Based on the anticipated subgoals, a continuous reward signal is provided to train a low-level, goal-conditioned policy under the hierarchical reinforcement learning (HRL) framework with provable sub-optimality bound. Extensive experiments on complex, long-horizon manipulation tasks show that ReLAM significantly accelerates learning and achieves superior performance compared to state-of-the-art methods.

## 1 Introduction

Reward design stands as one of the most fundamental challenges in reinforcement learning (RL), particularly in the domain of vision-based robotic manipulation (Tian et al., 2023; Lu et al., 2025; Escontrela et al., 2023; Huang et al., 2024; Pang et al., 2025). In simulated environments, a common and often effective approach is to engineer dense reward signals based on precise geometric information, such as the Euclidean distance between a robot's end-effector and a target position. However, this paradigm faces a critical limitation in real-world applications: exact state information is typically unavailable due to sensory noise, occlusions, and perceptual ambiguities. Consequently, agents must rely on high-dimensional visual observations, making hand-engineered reward design not only labor-intensive but also notoriously challenging. This reward specification bottleneck severely impedes the scalability and adoption of RL in practical robotic settings.

Some prior works overcome this limitation by adopting Learning from Observation (LfO) approaches. A common practice is to employ adversarial frameworks (Ho & Ermon, 2016; Torabi et al., 2018; Kostrikov et al., 2019), where a discriminator that does not take action as input is trained and subsequently used as a reward function. However, when dealing with high-dimensional visual inputs, such methods suffer from significant challenges in terms of training difficulty and stability. In recent years, several works (Tian et al., 2023; Sontakke et al., 2023; Ma et al., 2023; Escontrela et al., 2023; Huang et al., 2024) have instead attempted to design visual rewards based on heuristic strategies. These approaches either yield sparse rewards or lack an explicit structured learning process, making them inefficient for long-horizon tasks with extended periods of partial observability or complex dynamics. Thus, there still remains a need for a framework that can automatically synthesize informative, dense reward signals from readily available video demonstrations, while guiding the agent through a structured and geometrically grounded learning curriculum.

In this work, we introduce Reward Learning with Anticipation Model (ReLAM), a novel framework that automatically generates dense and structured rewards from action-free video demonstrations.

ReLAM is built on the recent insight that *object keypoints can serve as a powerful intermediate representation for capturing task geometry and progression* (Wen et al., 2024). ReLAM begins by extracting task-relevant keypoints from video demonstrations: we first use the Segment Anything Model (SAM) (Zhang et al., 2024) to isolate objects of interest, then apply a tracking model (Karaev et al., 2024b) to follow pixel-level features across frames. A sparse set of representative points is selected and propagated consistently, forming a trajectory of keypoints that encode object motion. From these, we identify keyframes that signify critical stages of the task, and define the keypoint configurations in those

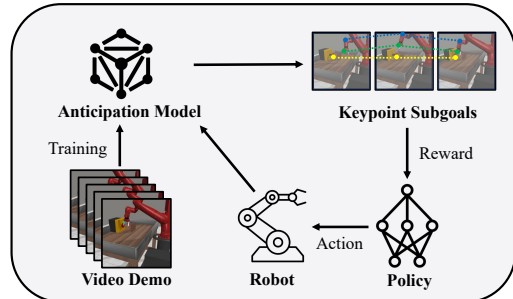

Figure 1: An illustration of ReLAM for generating the keypoint subgoals with anticipation model and calculating rewards for a goal-conditioned policy.

frames as subgoals. Using this curated dataset, ReLAM learns an anticipation model capable of predicting a sequence of intermediate keypoint-based subgoals that lead to the final goal. This model acts as a high-level planner, constructing a structured curriculum aligned with the geometric requirements of the task. The anticipated subgoals then enable the computation of a continuous reward signal based on keypoint distance, which is used to train a low-level, goal-conditioned policy under the hierarchical RL (HRL) framework with provable sub-optimality bound.

Our contributions are as follows: First, we make a novel derivation from the established point-to-point movement principle (Wen et al., 2024) specifically for reward design: we demonstrate that the distances between learned keypoints provide a meaningful reward signal. Second, we introduce ReLAM, a novel framework that uniquely combines this keypoint-based reward with an anticipative generative model to automatically construct a structured learning curriculum from mere video demonstrations, entirely without action labels. Third, our method bridges the gap between high-level planning and low-level control within an HRL framework, where the anticipation model proposes geometrically meaningful subgoals and the dense, keypoint-derived reward signal reliably guides policy optimization with provable sub-optimality bound. Finally, through extensive empirical validation, we demonstrate that this approach not only significantly accelerates learning but also achieves new state-of-the-art performance on long-horizon tasks, thereby offering a robust and practical pathway toward scalable visual reinforcement learning for robotics.

## 2 RELATED WORK

### 2.1 ROBOTIC MANIPULATION WITH VISUAL INPUT

Robotic Manipulation with visual input has long been a prominent research topic. Traditional approaches rely on supervised learning for behavior cloning, and this paradigm has continued to evolve, giving rise to methods such as Diffusion Policy (Chi et al., 2023; Ze et al., 2024) and VLA-based methods (Kim et al., 2024; Black et al., 2024). However, these approaches require large amounts of data and tend to suffer from substantial compounding errors in long-horizon tasks. In light of these issues, many studies have adopted reinforcement learning to train control policies based on visual input. For example, VPG (Zeng et al., 2018) and QT-Opt (Kalashnikov et al., 2018) apply the vision-based reinforcement learning framework to learn a grasping policy. Recent works (Ren et al., 2025; Lu et al., 2025) have employed RL to diffusion policy or VLA model, demonstrating promising performance in robotic manipulation tasks. Although these image-based reinforcement learning methods show considerable promise, they share a common challenge: the difficulty of reward design. Recently, ATM (Wen et al., 2024) abstracted images into a set of representative keypoints as task representations and employed behavior cloning to train policies, demonstrating strong generalization capability. Motivated by their method, we propose a keypoint-based reward learning approach, which provides an effective solution to the challenge of reward design for robotic manipulation.

## 2.2 Reward Learning from Videos

A common source of reward functions in visual reinforcement learning is the extraction of signals from videos, particularly from expert video demonstrations. Some adversarial imitation learning approaches (Li et al., 2017; Torabi et al., 2018; Rafailov et al., 2021; Kostrikov et al., 2019) employ the output of a discriminator as the reward function; however, such methods often exhibit instability when handling high-dimensional inputs. Benefiting from recent advances in foundation models, a number of works (Tian et al., 2023; Ma et al., 2023; Sontakke et al., 2023) instead use the distance between observations and target images/videos in the representation space as rewards. Since generated targets from generative models typically contain considerable noise and blurriness, these approaches usually require pre-given target images, which limits their applicability to open-ended tasks. To address the issue of inaccurate generation, some methods (Escontrela et al., 2023; Huang et al., 2024) indirectly leverage the model's confidence in its generated outputs as a reward signal. Such methods rely entirely on generative models and lack a substantive understanding of the spatial and temporal structures of the task. As a result, they continue to exhibit constraints in unseen areas. In contrast, ReLAM effectively extracts structural information across different dimensions of the task from video demonstrations, simplifying the task into point-to-point movements and thereby yielding more generalizable rewards.

## 2.3 Hierarchical Reinforcement Learning

Hierarchical reinforcement learning (HRL) aims to improve scalability and efficiency in long-horizon tasks by introducing temporal abstractions. Early frameworks such as Options (Sutton et al., 1999) and MAXQ (Dietterich, 2000) formalize sub-task structures through temporally extended actions and value function decomposition. More recent works focus on goal-conditioned hierarchies and often employ a high-level policy, which can be either a neural network (Nachum et al., 2018; Chane-Sane et al., 2021) or even some foundation models (Pang et al., 2023), to generate subgoals and a low-level policy to execute. It is argued that the high-level policy, which can be called as an anticipation model (Yu, 2025), should identify a waypoint that lies on an optimal shortest path to the final goal to find a global optimal policy. In this work, we will leverage the geometric priors inherent in robotic manipulation tasks to learn an anticipation model capable of continuously generating subgoals and train policy under the HRL framework.

## 3 Method

This section presents the method ReLAM which automatically provides reward by learning from video demonstrations $\mathcal{D} = \{V_i = (I_1^i, I_2^i, \cdots, I_{T_i}^i)\}_{i=1,\cdots,N}$. We divide our approach into two stages. The first stage is to learn an anticipation model which takes in the current task state and desired final goal as input to produce a relatively easy-to-reach intermediate keypoint-based subgoal. At the second stage, with the assistance of the anticipation model, a dense reward function is designed to train a low-level, goal-conditioned policy. We will elaborate on these two stages below.

### 3.1 Anticipation Model Learning with Keypoints

This part introduces how we learn an anticipation model from video demonstrations. Instead of training the anticipation model to generate images, we simplify it into a keypoint generation model like ATM (Wen et al., 2024). A good selection of keypoints can be a highly abstract and effective representation of the task, and will reduce the difficulty of generating subgoals simultaneously. In the following section, the learning procedure of keypoint-based anticipation model will be presented by answering three questions: (1) How to select the representative keypoints? (2) How to determine an appropiate subgoal for anticipation model to generate? (3) How to train the anticipation model?

#### 3.1.1 Subgoal Dataset Generation

**Keypoint Selection** For the first question, i.e., to select the representative keypoint in one image, it is important to first pick out the key objects. ATM (Wen et al., 2024) samples pixels averagely in one image, which might make too many points chosen, leaving these points unrepesentative. In ReLAM, we propose a new sampling strategy to elect the representative keypoints. First, for each

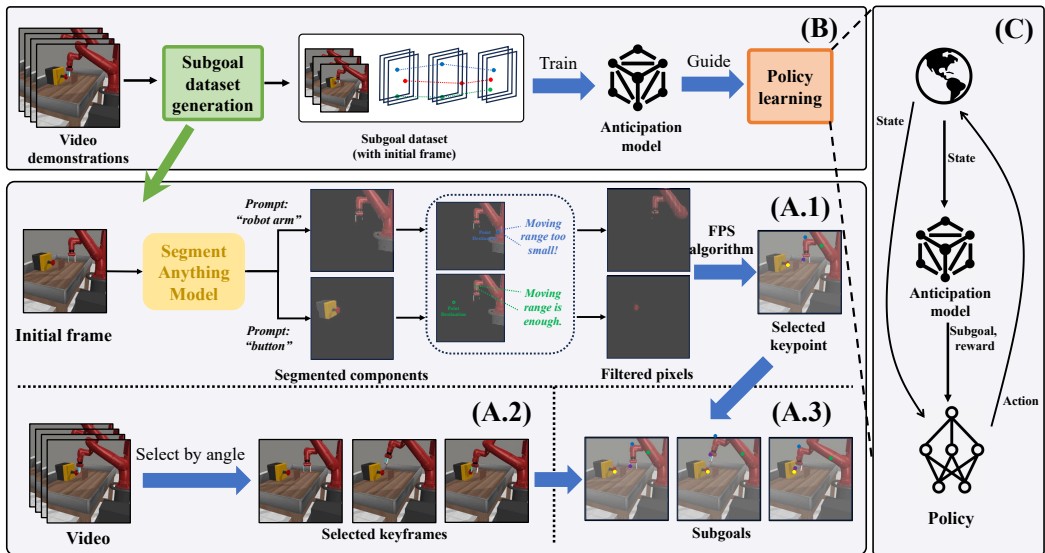

Figure 2: Overall training framework ReLAM method. (A) ReLAM first picks out representative keypoints in the initial frame of the video and then selects keyframes throught the video, turning the position of keypoints in these frames into subgoals. (B) Training the anticipation model based on the generated subgoal dataset. (C) Training policy with point-based reward with subgoals generated from anticipation model.

video demonstration, we extract its first frame and apply a grounded SAM model (Zhang et al., 2024) to obtain task-relevant segmentations. Next, for the pixels corresponding to each key object in the image, we employ a track model (Karaev et al., 2024a) to follow their motion trajectories throughout the entire video demonstration. Specifically, each trajectory records the two-dimensional coordinates of the pixel within the image coordinate system across all frames of the demonstration. Among all pixels corresponding to key objects in the image, we identify those that are truly relevant to the task by applying a predefined threshold to remove pixels whose motion range across the video is negligible. After filtering out pixels with small displacements, we further select the final keypoints using Farthest Point Sampling (FPS) (Eldar et al., 1997). The entire procedure for keypoint selection can be summarized by the following formulation:

$$\mathcal{P} = \textbf{FPS}\left(\{p = (x, y) \in SAM(I_0) : \max_{0 \le t, t' \le T}(x_t - x_{t'})^2 + (y_t - y_{t'})^2 \ge \Theta\}\right) \quad (1)$$

For the equation above, $I_0$ represents the initial frame of the video, $SAM$ denotes the segmentation model that picks out the task-relevant pixels, $(x_t, y_t)$ means to which position the point $(x, y)$ in $I_o$ will move at time $t$ in the video, $\Theta$ is a predefined threshold and $\textbf{FPS}$ denotes the Farthest Point Sampling technique. In most robotic manipulation tasks, these points serve as a high-level abstraction of the task state. By tracking the motion of these keypoints, one can infer the location and posture of the robotic arm, as well as whether it has performed the intended action on the object.

**Keyframe Selection** Filtering task-relevant keypoints in the image simplifies and condenses the spatial structure of the task. For a video sequence, however, the temporal dimension is equally important, as it reveals the underlying logic and patterns of the robot arm's motion, which can assist to determine which subgoal for the anticipation model to generate. Suppose a robotic arm is instructed to press a button with a wall obstructing between, directly using the final goal position of the arm as guidance, i.e. where the button places, might mislead the robot to collide with the wall. In such cases, the task is usually decomposed into two steps: first, moving around the wall, and second, pressing the button. Each step is relatively simple for the robotic arm, whereas executing them simultaneously as a single step would be considerably more challenging. To formalize this decomposition, we first introduce the following definition: a robot arm motion is said to be a *linear motion*, if the arm is able to move from the starting point to the target point along a straight line. Based on this concept, we posit that a robotic manipulation task can be decomposed into multiple segments, each representing a linear motion. Under this assumption, the frames situated between consecutive linear motions can be identified as keyframes. Extracting these keyframes enables us

to characterize the intrinsic motion regularities of the task. Combining these keyframes with the keypoints elected with Eq (1), the position of keypoints in these images become a perfect subgoal for the anticipation model to generate, which marks the optimal path to the final goal.

In ReLAM, keyframes are picked out from the video demonstration every certain interval. Specifically, we predefine a minimum step size $m$ and a maximum step size $M$. We then track the movements of the keypoints and, within the step range $[m, M]$, identify the timestep at which the change of keypoint motion is most pronounced. Specifically, since we assume that keyframes lie at the transition between two linear motions, we determine them based on the angle between the displacement vector of the current timestep and that of the previous timestep: if the frame lies within a linear motion, the angle is nearly zero; whereas at the boundary between two consecutive linear motions, the angle becomes significantly larger, in which case the frame is regarded as a keyframe. This keyframe selection process can be formalized as follows:

$$t_j = \arg \min_{t \in [t_{j-1}+m,\ t_{j-1}+M]} \sum_{k=1}^{K} \frac{\langle p_t^k - p_{t-1}^k, p_{t+1}^k - p_t^k \rangle}{\|p_t^k - p_{t-1}^k\| \|p_{t+1}^k - p_t^k\|} \tag{2}$$

where $t_j$ is the timestep for $j$-th keyframe, $p_t^k$ denotes the coordinate of the $k$-th keypoint at timestep $t$ and $\langle \cdot, \cdot \rangle$ is the inner product operation. For each video demo, we take the keypoints extracted from the initial frame using Eq. (1) and track their coordinates across the video keyframes obtained via Eq. (2). In this way, we construct the keypoint dataset below, with $p_i^k$ being the position of $k$-th keypoint at keyframe $j$ for demo $i$, and $x_{i,j}^k, y_{i,j}^k$ being its corresponding coordinate.

$$\mathcal{K} = \bigcup_{i=1}^{N} \mathcal{K}_i = \bigcup_{i=1}^{N} \{p_{i,j}^k = (x_{i,j}^k, y_{i,j}^k)\}$$

### 3.1.2 ANTICIPATION MODEL LEARNING

The dataset $\mathcal{K}$ can be regarded as a collection of subgoal sequences composed of keypoint coordinates. Therefore, we employ an autoregressive model as the anticipation model to generate these subgoals sequentially. The anticipation model takes the initial visual observation $I_0$ of the task as input and performs two steps: (1) it identifies the keypoints within $I_0$ and records their coordinates $P_0$; (2) based on $I_0$ and $P_0$, it autoregressively predicts the coordinates of these keypoints in the subsequent keyframes. Note that ReLAM can be extended to multi-task scenarios by adding a task indication frame $I_{\text{task}}$ to the anticipation model's input. This frame serves solely to identify the current task and *remains constant across all states within a task and can be predefined*.

For image inputs, previous research (Zhou et al., 2024) have shown that directly leveraging representations from pretrained vision models often endows the model with stronger spatial understanding, thereby enhancing its generalization capability. Motivated by this observation, we

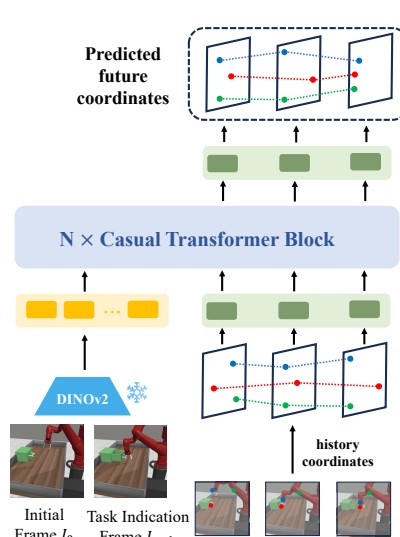

Figure 3: The structure of the anticipation model for subgoal generation.

also adopt a frozen DINOv2 (Oquab et al., 2023) model to extract image embeddings. The visual input for anticipation model here is two RGB images of size $256 \times 256$, one being initial frame and one being $I_{task}$. After being processed by the DINOv2 model, each image is divided into $16 \times 16$ patch embeddings. These patch embeddings are then concatenated with the tokens formed by the coordinates of keypoints from historical keyframes and fed into the model. The coordinates of the keypoints are first normalized and then mapped through a Multilayer Perceptron (MLP) into the embedding dimension. In the case of the first step, where no historical keypoints exist, we instead use a fixed special token to indicate that the model should predict the keypoint positions based on the given images. After being fed into the model, the image and point embeddings will pass through 12 layers of causal transformer blocks. Subsequently, the tokens corresponding to the points are processed by a MLP to predict the point coordinates in a residual form. These predicted coordinates are then compared with the ground-truth coordinates using Mean Squared Error loss under a

teacher-forcing scheme to train the anticipation model. The structure of the anticipation model is displayed in Fig. 3.

## 3.2 POLICY LEARNING WITH POINT-BASED REWARD

We train our policy under the hierarchical reinforcement learning framework. At the beginning of each episode, the initial image is fed into the anticipation model trained in the previous section. The model first predicts the keypoints' location $P_0$, and then autoregressively generates a sequence of subgoals (i.e., keypoint) $P_1, \cdots, P_k$. Our objective is to design a reward function that encourages $P_0$ to sequentially move towards $P_1, \cdots, P_k$, thereby enabling the robot arm to successfully complete the task. Based on the assumption that motion between keyframes is linear, the transitions from $P_j$ to $P_{j+1}$ correspond to approximately a straight path. Therefore, the reward can be directly defined using the Euclidean distance in the pixel coordinate system. Formally, we define the movement of a subgoal from $P_{j-1}$ to $P_j$ as the $j$-th stage. For this stage, the distance between the current position of the keypoint and the subgoal $P_j$ can be expressed as:

$$l = \frac{1}{K} \sum_{k=1}^{K} \|p^k - p_j^k\|_2 \tag{3}$$

where $p^k$ denotes the current position of the $k$-th keypoint, $p_j^k$ represents its target position at stage $j$. Next, a monotonic function is employed to transform the distance into dense reward $r_{\text{dense}}$. We find that a piecewise linear function performs best and the results can be seen in Fig. 6(b). We assume that when the distance between a keypoint and the subgoal is smaller than a predefined threshold $\theta_s$, the robot is considered to have successfully achieved the subgoal of stage $s$. At this point, the process transitions to stage $s + 1$, with the subgoal updated to the $(s + 1)$-th target position. Upon completing each subgoal, the robot receives an additional stage-success reward, and upon accomplishing the entire task, it is granted a final success reward. Consequently, the overall reward can be expressed as follows:

$$r = r_{\text{dense}} + r_{\text{success}} + I(l_s \leq \theta_s) \tag{4}$$

We find that when trained with this kind of reward, the policy is able to find a near-shortest path to complete the task. We provide a brief mathematical proof to show this near-optimality in Appx. E.

## 4 EXPERIMENTS

In this section, we conduct extensive experiments to evaluate our proposed method in robotic manipulation tasks. We first introduce the experiment setup.

### 4.1 EXPERIMENT SETUP

**Evaluation environments.** We conduct experiments on two robotics manipulation environments: Meta-World (Yu et al., 2019) and ManiSkill (Gu et al., 2023), as shown in Fig. 7. **(1) Meta-World:** This environment requires the agent to control a Sawyer robotics arm with 7 degrees of freedom (DoF) and a parallel finger gripper. Meta-World offers a suite of 50 distinct manipulation tasks, covering a wide array of scenarios, such as interactions with drawers, buttons, doors and balls. For our experiments, we assess the performance of our methods on a subset of tasks: drawer opening, door opening and button pressing. **(2) ManiSkill:** ManiSkill is a powerful unified framework for robot simulation and training powered by SAPIEN. Here we focus on table-top manipulation tasks, which involve a Panda robotic arm by Franka Emika with 7 DoF and a parallel finger gripper. These tasks are primarily focused on block manipulation tasks, which are designed to test the robot's foundational skills, such as reaching a goal point. We mainly use Drawer Open, Door Open and Button Press Wall from Meta-World, and Push Cube, Pick Cube from ManiSkill for evaluation. The observations on all tasks are images with $256 \times 256$ pixels, which are captured by the fixed-position third-person camera. We run online RL for Meta-World and offline RL for ManiSkill environments.

**Dataset for training. Video demonstration dataset** contains 100 trajectories collected by motion planning for each task in both Meta-World and ManiSkill. This video dataset is action-free and used to generate keypoint subgoal dataset for the training of anticipation model. Besides the video demo

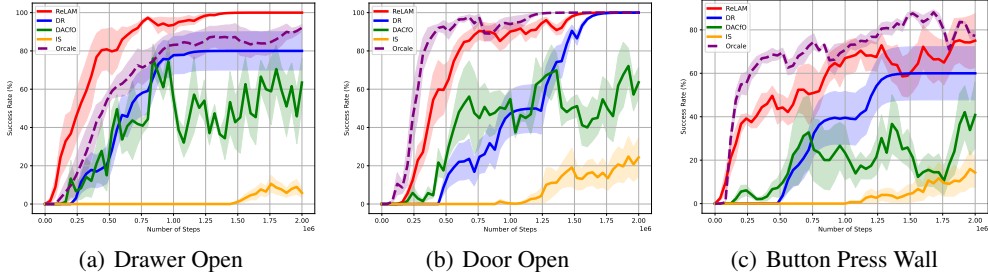

(a) Drawer Open          (b) Door Open          (c) Button Press Wall

Figure 4: Performance of different methods on Meta-World tasks. The x-axis denotes the number of interaction steps with the environment, and the y-axis denotes the average success rate, by evaluation for 30 episodes. The error bars stand for the half standard deviation over five seeds.

dataset, we also have an **offline control dataset** which contains action for offline reinforcement learning setting with 200 trajectories gathered for each task. Among these trajectories, 100 of them are expert demonstrations and another 100 are obtained by adding random noise to expert action.

**Implementation details.** For online RL with Meta-World, We build upon the well-established open-source reinforcement learning library Stable Baselines3 (Raffin et al., 2021), utilizing its PPO implementation. In some tasks, we slightly adjust the camera viewpoints to prevent severe occlusion of task-relevant objects. For offline RL setting on ManiSkill, we utilize OfflineRL-kit (Sun, 2023), a well-verified offline RL codebase. Specifically, we use Implicit Q-Learning (Kostrikov et al., 2022), an offline reinforcement learning algorithm that avoids explicit policy constraints by learning value functions implicitly and extracting a policy through advantage-weighted regression.

## 4.2 MAIN RESULTS

**Baselines for comparison** We choose the following representative approaches which learn a reward from videos for comparison. (1) **DACfO** is an adversarial imitation learning method which combines the idea of DAC (Kostrikov et al., 2019) and GAIfO (Torabi et al., 2018), where we modify the discriminator's input like GAIfO to consist of the current observation $o$ and the next observation $o'$, enabling it to handle action-free demonstration datasets. For offline data, we first run DACfO online and save the last 10 checkpoints of the discriminator. Then we use them to label the offline dataset with ensemble technique. (2) **Diffusion Reward (DR)** (Chi et al., 2023) trains a diffusion model with the video demo data and learn policy by computing the conditional entropy of the diffusion model as reward. (3) **Image Subgoal (IS)** integrates our method with the idea of VP2 (Tian et al., 2023) by employing a flow matching model to autoregressively generate subgoals from the initial image, and then uses the cosine similarity between the current visual observation and the target image in the representation space of DINOv2 (Oquab et al., 2023) as the reward to train a goal-conditioned policy with image subgoal. (4) **Orcale** replaces the generated image subgoals in IS baseline with the ground-truth ones for each episode, with all other components unchanged.

**Results for Meta-World.** Fig. 4(a), 4(b), 4(c) shows the success rate of different reward learning methods for online reinforcement learning results in Metaworld environments. In general, our proposed method ReLAM outperforms the baselines for all three environments. It can be observed that on these tasks, ReLAM rapidly achieves very high success rate. In contrast, other baseline methods either fail to reach such high success rates or require significantly more interaction steps. We evaluate for five fixed seeds ($0 - 4$), and it is worth noting that for some seeds, the Diffusion Reward approach completely fails to learn. This occurs because their method relies on an auxiliary RND reward to encourage exploration, which does not necessarily provide a correct exploration signal and instead results in highly stochastic exploration. Under such circumstances, certain seeds may never encounter the correct trajectory, ultimately preventing successful learning. In contrast, our method incorporates both the current and target coordinates of keypoints as part of the policy input, inherently providing the policy with implicit guidance. Moreover, the distance-based reward enables the policy to gradually recognize that approaching the target yields higher returns, thereby steering exploration toward meaningful regions of the state space and allowing the agent to acquire the task more efficiently. For DACfO, the curve exhibits substantial fluctuations, indicating that the training process is indeed unstable. For Orcale, since it leverages ground-truth subgoal images, we can see that learning proceeds relatively quickly and ultimately achieves a high success rate. In contrast, when we replace the subgoals with results generated by the flow matching model, the performance, as shown by IS, drops significantly. The generated images often contain noise and

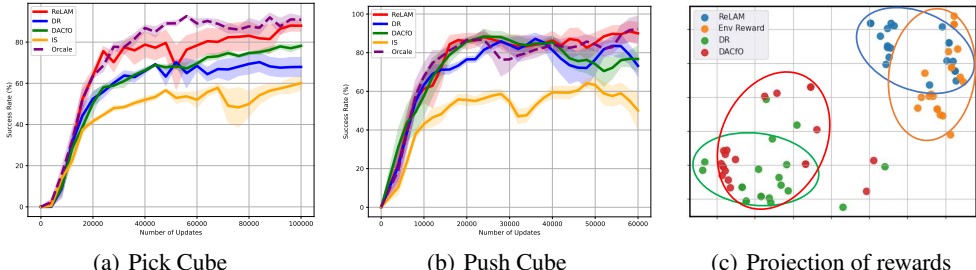

(a) Pick Cube      (b) Push Cube      (c) Projection of rewards

Figure 5: (a), (b): Performance of different methods on ManiSkill tasks. The x-axis denotes the number of update, and the y-axis denotes the average success rate, by evaluation for 30 episodes. The error bars stand for the half standard deviation over five seeds. (c): t-SNE projections of the rewards generated by different methods.

local blurriness, making it difficult to establish a consistent similarity threshold in the representation space. For example, while a threshold of 0.95 may be appropriate for the first generated result, the second might require 0.9, and this threshold can vary further depending on the initial state of the task. Due to this inconsistency, the IS method achieves very low success rates, with only a small fraction of well-generated cases for the policy to learn.

**Results for ManiSkill.** Fig. 5(a), 5(b) displays the success rate of different approaches for offline reinforcement learning results in ManiSkill environments. On Pick Cube and Push Cube tasks, our method surpasses all baseline approaches except Orcale which cheats with the ground-truth image subgoal. It can be observed that ReLAM achieves performance comparable to Orcale, whereas IS shows a clear performance drop compared to Orcale. This indicates that even without access to privileged information, by leveraging abstract keypoints as targets, ReLAM not only reduces the generation difficulty for the anticipation model but also effectively captures structural information of the task to guide the policy. The reason why IS achieves much higher success rates than Meta-World is that: (1) The offline control dataset contains expert action label, reducing the need for exploration; (2) The sequence of subgoals for ManiSkill is shorter, leading to less compounding error of the generation results anticipation model. The performance of Diffusion Reward and DACfO is also quite similar, as both methods share essentially the same underlying principle: rewards are higher near the expert distribution and lower when further away from it. Since a large portion of the offline decision-making data is near-expert, the rewards assigned by both methods are generally high, which explains why they ultimately achieve comparable results. However, their performances still fall short of ReLAM . We think this is because our approach can recognize and reward trajectories that deviate slightly from the expert distribution yet still move effectively toward the goal. This is enabled by our distance-based reward design, which provides the policy with a strong guidance signal and fosters a deeper understanding of the task's spatiotemporal structure. In contrast, Diffusion Reward may assign lower returns to such trajectories due to its higher entropy, leading to less efficient learning.

We make a visualization of the rewards for different methods in Fig. 5(c), which verifies our analysis above. We sample 20 trajectory segments for Pick Cube task and label them with four types of rewards: environment rewards, ReLAM, DR, and DACfO. We then projected the labeled trajectories into a two-dimensional space using t-SNE, where each trajectory corresponds to a single point. Since ReLAM assigns rewards based on keypoint distances and the environment reward is based on distances in the world coordinate system, these two rewards are relatively close. Moreover, as mentioned above, both Diffusion Reward and DACfO assign higher rewards to regions closer to the expert distribution, which explains why their projections are not far away from each other.

### 4.3 ABLATION STUDY

**Effect of the point number.** We study whether the number of points selected for the task will affect the performance of the policy. We sample 4, 8 and 12 points for Drawer Open task and the result is shown in Fig. 6(a). The best performance is achieved when sampling four keypoints, followed by eight keypoints, while twelve keypoints yield the weakest results. This outcome can be explained by the truth that for Drawer Open task, sampling four keypoints—three on the robotic arm and one on the drawer—is sufficient to provide an adequate representation of the task state, thereby enabling rapid policy learning. In contrast, with eight or twelve keypoints, the prediction difficulty for the anticipation model increases. Moreover, requiring the policy to simultaneously drive all keypoints

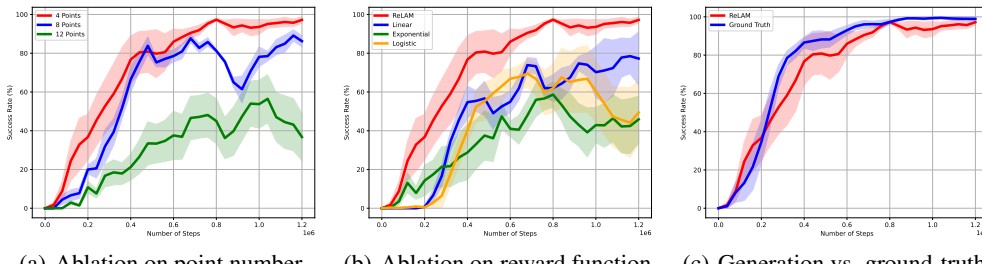

(a) Ablation on point number    (b) Ablation on reward function    (c) Generation vs. ground-truth

Figure 6: Ablation study of ReLAM on Drawer Open task.

toward their respective targets imposes stricter constraints on its actions, effectively forcing the behavior to closely mimic the demonstrations. In this regard, four points can provide a sufficiently broad criterion while allowing more flexible exploration.

**Effect of the reward function.** We evaluate the impact of different type of reward functions conditioned on the keypoint distance have on the performance of reinforcement learning. We introduce three kinds of reward functions and compare them to the separate linear function used in ReLAM: (1) pure linear function; (2) exponential function; (3) logistic function. All these functions are designed to have the same range when the point distance is between $[0, 30]$. The performance is displayed in Fig. 6(b). We found that the piecewise linear function took the lead and linear function achieved relatively good performance, while the exponential and logarithmic functions performed worse. We think this is because the slopes of the latter two functions vary continuously, making it difficult for the policy to adapt; while the slope of the linear function remains constant, resulting in insufficient encouragement for the policy as it approaches the target. The piecewise linear function, however, strikes a balance between the two: it provides sufficient incentives for the policy to reach the goal while maintaining a certain degree of stability.

**Generated subgoal versus ground-truth subgoal.** To evaluate the accuracy of the anticipation model in generating subgoals, we compare its predictions with the ground-truth subgoals. Specifically, at each environment initialization, we provide the subsequent sequence of ground-truth keypoint subgoals and train a goal-conditioned RL policy based on these targets like ReLAM. Fig. 6(c) presents a comparison between the performance of policies trained with AR-generated subgoals and those trained with ground-truth subgoals. The performance gap between the two is relatively small, indicating that the anticipation model produces sufficiently accurate subgoals. In contrast, when the subgoals are represented as images rather than points, the performance of Image Subgoal drops significantly compared to the Orcale baseline. This result further confirms that point-based representations substantially reduce the difficulty of the generation problem, thereby enabling point-based rewards to effectively guide the policy toward task completion.

## 5 CONCLUSION

This study explores the reward design problem for robotic manipulation. We propose a novel approach, ReLAM, which first learns an anticipation model that serves as a planner and proposes intermediate keypoint-based subgoals and then train a goal-conditioned policy with the distance of keypoints as reward signal. We conduct extensive experiments and demonstrate that ReLAM is capable of being applied to a variety of robotic platforms, enabling a robust and practical pathway towards scalable RL for robotic manipulation. Despite the promising results, there are still limitations. One limitation of our work is reliance on the viewpoint. Our anticipation model is trained with video demos from a single camera with the assistance of a track model. If the viewpoint undergoes a dramatic change, the model will struggle to generate the desired target. Moreover, significant occlusions can prevent the track model from accurately following the keypoints. A potential solution is to use observation from multi views and merge them into the point cloud, which is more robust to viewpoint disturbance and occlusion. Besides, the experiment scale is limited, in terms of the dataset scale and model size. In future works, we hope to scale up the framework to solve more challenging tasks. For instance, employing a pre-trained VLM as the anticipation model such as Qwen-VL-2.5 (Bai et al., 2025), and training with more data like Open X-Embodiment (Collaboration, 2023). We believe these interesting directions are worth further exploration for developing smarter and more robust robots with the support of more general-purpose reward and reinforcement learning.

ETHICS STATEMENT

In the development and evaluation of ReLAM for robotic manipulation, we have carefully considered the ethical implications of this research, particularly as they pertain to the use of robotic manipulation tasks and artificial intelligence. The proposed involves the use of some free open-source vision foundation models, along with the collection and use of data in simulators. ReLAM is designed to respect privacy and ensure the security of these models and data. The datasets used do not contain any personal or sensitive information, and all data collection processes comply with relevant legal standards and best practices in research ethics. The potential for bias and discrimination has been addressed by ensuring that the anticipation model does not inadvertently generate any biased results of the environment. This is particularly important in maintaining fairness and avoiding any form of discrimination that could arise from biased training data. The research has been conducted with a commitment to research integrity, including thorough documentation and adherence to IRB guidelines where applicable. We recognize that the insights and methods presented in this paper must be applied responsibly, avoiding any potentially harmful applications. The technology developed is intended for beneficial purposes and should not be used in ways that could cause harm or diminish the safety of individuals. The experiment results are reported with the most transparency and accuracy, reflecting our commitment to advancing knowledge in the field of robotic manipulation while upholding the highest ethical standards.

REPRODUCIBILITY STATEMENT

In this study, to ensure the reproducibility of our approach, we provide key information from our submission as follows.

1. **Training Algorithm.** We provide our approach in Sec. 3.
2. **Experimental Details.** We list the detailed experiment settings in Sec. 4.1, Appx. A and hyperparameters in Appx. C.
3. **Derivation Details.** We provide the missing proofs in Appx. E.

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

# Appendix

## ACKNOWLEDGMENT FOR LLM USAGE

We acknowledge that the Large Language Model was employed solely for polishing the language of certain paragraphs in this manuscript. The model was not used for any other part of the work. All scientific contributions, including conceptualization, method, experimentation and analysis, are entirely the work of the authors.

## A  MORE IMPLEMENTATION DETAILS & EXPERIMENT SETUP

### A.1  MORE DETAILS FOR LEARNING ANTICIPATION MODEL

After obtaining the segmentation of task-relevant objects in the keypoint selection stage, we perform an additional filtering step: a point is retained only if it, along with all the pixels within an $L \times L$ square centered on it, lies inside the segmentation. This is because the SAM model sometimes includes extra pixels from the background or other irrelevant objects, which usually appear as isolated rather than contiguous regions. The above operation effectively filters out such points.

As for the track model, It is important to note that it is capable of following points that initially appear within the image boundaries but later move outside the frame. In other words, assuming the image size is $H \times W$, the coordinates of a point $(x_t, y_t)$ may take values such as $x_t < 0$ or $y_t > W$. This property further ensures the model's robustness in tracking keypoints and extends its effective tracking range.

### A.2  MORE DETAILS FOR POLICY LEARNING

As we say, various monotonic functions can be used to form a keypoint distance-based reward, such as exponential, logarithmic, or linear functions. But We find that a piecewise linear function performs best. We provide its specific form below:

$$
\begin{cases}
r_{\text{dense}} = k_s \cdot (l - l_s) + b_s, & l_s \leq l \leq l_{s+1}, \\
k_s = \dfrac{b_{s+1} - b_s}{l_{s+1} - l_s}.
\end{cases}
\tag{5}
$$

We ensure that $k_s > k_{s+1}$, meaning that as the keypoint approaches the target position, the slope of the reward function gradually increases. This gives continuous and stable encouragement to the policy to reach the desired subgoal. During inference, we do as the training stage: first generate the subgoal sequences with anticiaption model, then instruct the policy to complete them one by one.

### A.3  MORE DETAILS FOR PPO

We make some modifications based on the source code of PPO in Stable Baselines3. To make the collecting process compatible with hierarchical reinforcement learning framework, we set $terminal = True$ once a subgoal is achieved or a whole episode ends. This operation segments the whole trajctory by subgoal, which makes GAE computation done separately for each low-level policy. Besides, we add reward scaling technique to make learning faster and more stable. For policy and critic network, the input consists of the current RGB observation $I$, the current coordinate of the keypoints $p$, and the target coordiante of the keypoints $p'$ predicted by anticipation model. For image $I$, we utilize a three-layer CNN network as encoder. For the points $p$ and $p'$, we flatten and feed them into a MLP to get point representation. The image and point features are concatenated together and sent into another MLP to get the final output.

### A.4 More Details for Offline RL

For offline RL with ReLAM , we first generate the desired subgoals using anticipation model for each trajectory. Then a similar way to online RL is employed to label the reward with distance like Eq. 4, and we will proceed to the next subgoal if the distance is within the threshold. For IS and Orcale baseline, we assign rewards in the same way to ReLAM. For DACfO, we first run online DAC in the task environment and save the last 10 checkpoints of the discriminator. After online training, rewards are given by the average of the output from these 10 discrimators, which is found to be more generalizable than using only one discrimator (Luo et al., 2022).

### A.5 Experiment environments

We provide a visualization of the experiment environments in Fig. 7.

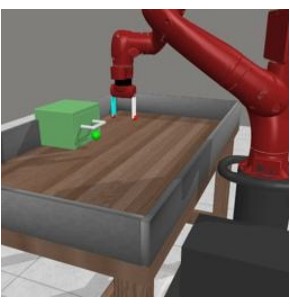 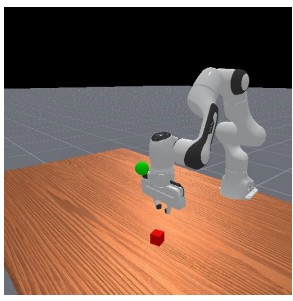

**(A) Meta-world**          **(B) ManiSkill**

Figure 7: A visualization of the environments in our experiments. **(A)** In Meta-world, the agent controls a Sawyer robot to manipulate various objects such as window, drawer and door. **(B)** In the ManiSkill environment, the agent controls a Franka robot with 7-DoF.

---

**Algorithm 1** Reward Learning with Anticipation Model (ReLAM)

---

**Required**: an action-free video demo dataset $\mathcal{D}$, a text-grounded SAM model EVF-SAM, an off-the-shelf track model Cotracker
**Output**: the optimized robotic control policy $\pi$.

1: Initialize the anticipation model $G_\phi$, policy $\pi_\Phi$, where the subscript denotes their parameters.
2: // Generate keypoint subgoal dataset $\mathcal{K}$
3: **for** each trajectory in $\mathcal{D}$ **do**
4:     Pick out the keypoint in the initial frame with Eq. (1).
5:     Select the key frames using Eq. (2).
6:     Generate the subgoal data with the coordinate of keypoints in key frames.
7: **end for**
8: // Training anticipation model
9: **while** training not converge **do**
10:     Sample keypoint subgoal data from $\mathcal{K}$.
11:     Update $\phi$ by predicting the keypoint sequences with teacher-forcing.
12: **end while**
13: // Training policy
14: **while** policy training not converge **do**
15:     Collect trajectories $(o_t, a_t, o_{t+1}, g_t)$ by rolling out $\pi_\Phi$.
16:     Compute reward for each transition using Eq. (4)
17:     Update $\pi_\Phi$ using collected trajectories with PPO.
18: **end while**
19: **return** the optimized policy $\pi$.

---

# B  ALGORITHM DESCRIPTION

The practical implementation of ReLAM method for online reinforcement learning is presented in the form of pseudo-code in Algorithm 1.

# C  HYPER PARAMETERS

Table 1: Core Hyper-parameters for Learning Anticipation Model

| Hyper-parameters | Value |
|---|---|
| Embedding dimension | 512 |
| Layer Num. | 12 |
| Dropout rate | 0.1 |
| Head Num. | 8 |
| Keypoint Num. | 4 |
| Batch size | 8 |
| Learning rate | $3 \times 10^{-5}$ |

Table 2: Core Hyper-parameters for PPO

| Hyper-parameters | Value |
|---|---|
| Learning rate | $3 \times 10^{-4}$ |
| Batch size | 64 |
| Number of epochs | 10 |
| Gamma | 0.99 |
| GAE lambda | 0.95 |
| Number of Steps | 2000 |
| Clip range | 0.2 |
| Entropy coefficient | 0.0 |
| Value function coefficient | 0.5 |
| Max gradient norm | 0.5 |
| CNN channels | $[16, 32, 64]$ |
| CNN kernal sizes | $[8, 8, 8]$ |
| CNN strides | $[4, 4, 4]$ |
| Mlp hidden dims | $[512, 256]$ |

Table 3: Core Hyper-parameters for IQL

| Hyper-parameters | Value |
|---|---|
| Learning rate | $3 \times 10^{-4}$ |
| Batch size | 64 |
| Step per epoch | 2000 |
| Number of epochs | 50 |
| Gamma | 0.99 |
| Tau | 0.005 |
| Expectile | 0.7 |
| Temperature | 3.0 |
| CNN channels | $[16, 32, 64]$ |
| CNN kernal sizes | $[8, 8, 8]$ |
| CNN strides | $[4, 4, 4]$ |
| Mlp hidden dims | $[256, 256]$ |

## D    PROMPT USED IN SAM

The prompts used in SAM model are listed below:

- **Button Press Wall**:
  - robot arm
  - button
- **Door Open**:
  - robot arm
  - door
- **Drawer Open**:
  - robot arm
  - green drawer
- **Push Cube**:
  - robot arm
  - blue cube
- **Pick Cube**:
  - robot arm
  - red cube

## E    MATHEMATICAL ANALYSIS

We provide a brief mathematical analysis on the effectiveness of ReLAM below. We start by providing an assumption about the learning environment.

**Assumption 1.** *We assume the learning environment, which takes the coordinate of keypoints in the image as state space, satisfies the following conditions:*

*(1) The state space $\mathcal{S}$ is continuous.*

*(2) The state space $\mathcal{S}$ has a Euclidean distance metric $d(\cdot, \cdot)$.*

*(3) The environment's transition function $P$ is deterministic.*

This assumption usually holds in the point space. Apart from this assumption, we impose an additional one that restricts the point's stepwise movement range.

**Assumption 2.** *For each timestep, the robot takes action $a$, and the point $s$ transits to $s'$, which satisfies the following condition:*

*(1) $s'$ is reachable.*

*(2) There exists a fixed constant $M$, s.t. $d(s, s') \leq M$.*

*(3) $\forall \hat{s} \in B(s, M)$, if $\hat{s}$ is reachable, there exists a unique action $\hat{a}$, s.t. $P(s, \hat{a}) = \hat{s}$. For simplicity, we denote $P^{-1}(s, \hat{s}) = \hat{a}$ as the inverse dynamics model.*

Assumption 2 can hold for robotic manipulation environments, especially when the action mode is set to be delta position of the end effector. Next, we will give a definition which describes the linear motion of robotic manipulation.

**Definition 1** (Linear Reachability). *We say $s'$ is linearly reachable from $s$, if: $\exists \epsilon > 0, \forall \hat{s} \in \mathcal{S}$, if $\exists t \in [0, 1]$, $d(\hat{s}, t \cdot s + (1 - t) \cdot s') < \epsilon$, then $\hat{s}$ is reachable.*

And we define two kinds of reward:

**Definition 2** (Reward definition). *Suppose the final goal is $g$, the we define two reward:*

*(1) Time reward:*

$$r_t(s, g) = \begin{cases} -1, & \text{if } g \text{ is not reached,} \\ 0, & \text{if } g \text{ is reached} \end{cases}$$

*(2) Distance reward:*

$$r_d(s, g) = \begin{cases} f(d(s, g)), & \text{if } g \text{ is not reached,} \\ 0, & \text{if } g \text{ is reached} \end{cases}$$

where $f$ is a monotonically decreasing function that is always negative.

With these definitions, we can obtain the following lemma:

**Lemma 1.** *Suppose we have a task starting from $s_0$, and the terminal point state is $g$. The environment satisfies assumption 1, 2. If $g$ is linearly reachable from $s_0$, then the optimal policy $r_d$ is also optimal for $r_t$.*

*Proof.* Based on assumption 2 and linear reachability, it is easy to find that one of the optimal policy for time reward $r_t$ is:

$$\pi_{r_t}^*(a|s) = \begin{cases} P^{-1}(s, s + \frac{M \cdot (g-s)}{d(s,g)}), & \text{if } d(s, g) > M, \\ P^{-1}(s, g), & \text{if } d(s, g) \leq M \end{cases} \tag{6}$$

For the distance reward $r_d$, when starting from $s_0$, suppose the episode ends at time $T$, then the total return $G$ is computed as:

$$\begin{aligned} G &= \sum_{t=0}^{T} r_d^t(s_t, g) \\ &\leq \sum_{t=0}^{T} f(d(s_t, g)) \\ &= \sum_{t=0}^{\lceil d(s,g)/M \rceil} f(d(s_t, g)) + \sum_{t=\lceil d(s,g)/M \rceil + 1}^{T} f(d(s_t, g)) \\ &\leq \sum_{t=0}^{\lceil d(s,g)/M \rceil} f(d(s_t, g)) + \sum_{t=\lceil d(s,g)/M \rceil + 1}^{T} 0 \\ &\leq \sum_{t=0}^{\lceil d(s,g)/M \rceil} f(d(\hat{s}_t, g)) \end{aligned} \tag{7}$$

where $\hat{s}_t = s_0 + t \cdot \frac{M \cdot (g-s_0)}{d(s_0,g)}$ lines in the straight line between $s_0$ and $g$. Eq. (7) shows that the optimal policy for distance reward $r_d$ is the same as Eq. (6). □

Obviously, the policy in Eq. (7) always takes the action to reach the farthest reachable point in the straight line between the current state and the desired goal if it is linearly reachable. Therefore, it defines a shortest path from the start point to the goal point. Next, we will introduce some propositions about the anticipation model and pixel tracking in robotic manipulation.

**Definition 3.** *We say $(g_0, g_1, \cdots, g_k)$ is a path for the start point state $s_0$ and the goal point state $g$, if $g_0 = s_0$, $g_k = g$ and $\forall 0 \leq i \leq k-1$, $g_{i+1}$ is linearly reachable from $g_i$. A path is said to be the shortest if $(g_0, \cdots, g_k) = \arg\min \sum_{i=0}^{k-1} d(g_i, g_{i+1})$ among all paths for the task.*

**Assumption 3.** *The learning task has a shortest path $(g_0, g_1, \cdots, g_k)$. And the anticipation model can predict a path $(\hat{g}_0, \cdots, \hat{g}_k)$ satisfying:*

$$\forall 0 \leq i \leq k, \quad d(\hat{g}_i, g_i) < \epsilon_A \tag{8}$$

With the assumption above, we get $k$ subtasks where for each subtask $i$, the low-level policy $\pi_i^{ReLAM}$ is trained with distance reward using PPO. By denoting the expected return of $\pi_i^{ReLAM}$ under a reward $r$ in subtask $i$ as $\hat{V}_{i,r}^{\pi_i^{ReLAM}}$, we can obtain the following theorem:

**Theorem 1** (sub-optimality bound for ReLAM). *If for all $0 \leq i \leq k - 1$, the low-level policy $\pi_i^{ReLAM}$ trained with reward $r^{ReLAM}$ satisfys:*

$$|V_{i,r_t}^{\pi_{i,r}^{*ReLAM}} - V_{i,r_t}^{\pi_i^{ReLAM}}| < \epsilon_\pi \tag{9}$$

*which means that the expected time used for $\pi_i^{ReLAM}$ and $\pi_{i,r^{ReLAM}}^*$ to complete the subtask $i$ is almost the same. Then ReLAM will train a policy $\pi^{ReLAM}$, whose sub-optimality is bounded by:*

$$V_{r_t}^* - V_{r_t}^{\pi^{ReLAM}} \leq k \cdot (\epsilon_\pi + \frac{2\epsilon_A}{M}) \tag{10}$$

*Proof.* For all $0 \leq i \leq k - 1$, we have:

$$\begin{aligned}
V_{i,r_t}^{\pi^{ReLAM}} &= V_{i,r_t}^{\pi^{ReLAM}} - V_{i,r_t}^{\pi_{i,r}^{*ReLAM}} + V_{i,r_t}^{\pi_{i,r}^{*ReLAM}} \\
&\geq -\epsilon_\pi + V_{i,r_t}^{\pi_{i,r}^{*ReLAM}} \\
&= -\epsilon_\pi + V_{i,r_t}^* \\
&= -\epsilon_\pi + \frac{d(\hat{g}_i, \hat{g}_{i+1})}{M} \\
&\geq -\epsilon_\pi + \frac{d(g_i, g_{i+1}) - 2\epsilon_A}{M}
\end{aligned} \tag{11}$$

The first inequality can be deduced by lemma 1 and the fact that $V_{i,r_t}^{\pi_{i,r}^{*ReLAM}} = V_{i,r_t}^{\pi_{r_t}^*} = V_{i,r_t}^*$. The penultimate line is from the definition of time reward, the max-step-size assumption 2 and the fact that $\hat{g}_{i+1}$ is linearly reachable from $\hat{g}_i$. The last inequality comes from assumption 3 and the triangle inequality. By summing from $i = 0$ to $k - 1$, we have:

$$\begin{aligned}
V_{r_t}^{\pi^{ReLAM}} &= \sum_{i=0}^{k-1} V_{i,r_t}^{\pi^{ReLAM}} \\
&\geq \sum_{i=0}^{k-1} \left[ -\epsilon_\pi + \frac{d(g_i, g_{i+1}) - 2\epsilon_A}{M} \right] \\
&= \frac{\sum_{i=0}^{k-1} d(g_i, g_{i+1})}{M} - k \cdot (\epsilon_\pi + \frac{2\epsilon_A}{M}) \\
&= V_{r_t}^* - k \cdot (\epsilon_\pi + \frac{2\epsilon_A}{M})
\end{aligned} \tag{12}$$

The last equality is because $(g_0, g_1, \cdots, g_k)$ is the shortest path. And finally we can get Eq. (10), which completes the proof. $\square$

Theorem 1 shows that the learnt policy $\pi^{ReLAM}$ will find an almost shortest path, validating the soundness of our approach.

## F    MORE EXPERIMENT RESULTS

We provide the mean success rate of the last evaluation for each methods on training tasks in Tab. 4.

|  | Diffusion Reward | DACfo | IS | Oracle | ReLAM |
|---|---|---|---|---|---|
| *Meta-World Environments* | | | | | |
| Drawer Open | 80.0 | 71.3 | 4.0 | 93.3 | 100.0 |
| Door Open | 100.0 | 70.0 | 24.7 | 100.0 | 100.0 |
| Button Press Wall | 60.0 | 47.8 | 12.7 | 76.7 | 75.8 |
| *ManiSkill Environments* | | | | | |
| Push Cube | 69.3 | 76.7 | 46.7 | 92.7 | 89.3 |
| Pick Cube | 68.0 | 78.7 | 60.7 | 90.7 | 88.0 |

Table 4: Mean success rate of the last evaluation for each methods on training tasks.

