# OpenReview forum: "ReLAM: Learning Anticipation Model for Rewarding Visual Robotic Manipulation"
_ICLR.cc/2026/Conference — ICLR 2026 Conference Withdrawn Submission_

### Official Review · Reviewer_WUVe · 2025-10-31

**Soundness:** 2
**Presentation:** 2
**Contribution:** 2
**Rating:** 2
**Confidence:** 5

**Summary:**

This paper introduces ReLAM. where an anticipation model is learned from video information alone to auto-regressively generate a sequence of subgoals. The subgoal sequence is then used for a generic subgoal following reward function that can be applied to any task that has reasonable subgoals. The paper makes an assumption about tasks having linear motions, which enables them to generate more accurate keyframes representing significant changes in direction of objects in a task. The keyframes generated from action-free videos serve as a dataset to train the anticipation model, which is then leveraged for simulation RL training of task-specific policies.

**Strengths:**

- The paper improves on previous work such as ATM by adding an additional filtering procedure to select more representative keypoints via a motion range filtering + farthest point sampling procedure. This removes possibly unnecessary keypoints from being tracked, which saves both memory and improves training performance of the anticipation model and downstream RL training from following the predicted subgoals.
- Under the assumption of linear motions the equation used to select keyframes (by minimizing the angle between keypoints) is quite interesting and looks fairly novel. The paper shows that this achieves its intended effect as the trained anticipation model is capable of solving the selected tasks.

**Weaknesses:**

- The assumption that there are only linear motions in a task is a rather strong assumption and one can imagine this also runs counter to the point of using RL in real/simulation. Plenty of tasks require non-linear motions that could still have some kind of key point representation that is non-intuitive to humans (e.g. a Push T [1] type task).
- The 2D point keypoint representation is fairly limiting. There are also plenty of tasks with a 2D point representation in image space would be insufficient for tracking motion since some tasks might have camera angles where an object looks still in 2D image space, but is in fact moving away from the camera slowly.
- While keypoints are interesting, they appear to be quite dense and would suffer significantly from occlusions. The authors themselves mention that they have to modify the default camera angles in section 4.1 implementation details. Note that with the original dense reward functions and/or demonstration datasets the selected tasks in MetaWorld and ManiSkill are all solvable with dense reward RL or offline+online RL (e.g. RLPD [2]).
- While there are ablation studies, I feel they are not very in-depth enough to really understand why e.g. less points for subgoal representation works better. It would have been interesting to analyze e.g. the value function or gradient norms of PPO during training, which paint a better picture of why some settings are better than others. Specfically for the subgoal keypoint number, it would have also been useful to see what happens with a single keypoint.
- The choice of tasks tested on seem overly simple. PickCube and PushCube are the 2 simplest tasks in ManiSkill and past work has solved those tasks with sparse rewards + demonstration data before. PushCube specifically is also solvable quite quickly with PPO + sparse rewards in the ManiSkill 3 GPU sim. Would be interesting to see this work tested on the more difficult metaworld and maniskill tasks (e.g. PegInsertionSide).


[1] https://maniskill.readthedocs.io/en/latest/tasks/table_top_gripper/#pusht-v1 - example video of task
[2] Ball et. al, "Efficient Online Reinforcement Learning with Offline Data", ICML 2023

**Questions:**

- What are the ground truth sub-goals mentioned in 4.3? Where are those from?
- How come PPO is only being run for metaworld and not maniskill? Specifically it looks like ManiSkill 3 was used in the paper which has PPO baselines and GPU parallelization meaning PPO runs quite fast. (This was determined by looking at the appendix images, I will note it seems authors mistakenly cited ManiSkill 2 which is the old non-gpu-parallelized version). Simultaneously, how come offline RL is not being run on metaworld, there are many strong offline RL baselines in metaworld that use a small dataset like RLPD [2]

Other:
- figures 4 a-c have a typo "Orcale", same for line 359

[2] Ball et. al, "Efficient Online Reinforcement Learning with Offline Data", ICML 2023

---

### Official Review · Reviewer_NNby · 2025-10-31

**Soundness:** 2
**Presentation:** 3
**Contribution:** 2
**Rating:** 4
**Confidence:** 4

**Summary:**

Designing reward functions for RL is challenging in part because the design space is huge, and in part because reward functions that are effective for policy learning often rely on privileged information that may not be available outside of hand-crafted simulation environments. This paper aims to learn a shaped reward function from a demonstration dataset for the target task(s) and then subsequently use the learned reward function for policy learning in online or offline RL training pipelines. A key contribution of this work is the manner in which the reward function is learned: the proposed method, ReLAM, segments objects from visual observations using SAM and then tracks the objects throughout the demonstration using a pixel-level tracking model. This process involves some manual labor (prompts for SAM) but otherwise relies on heuristics for e.g. keypoint and keyframe selection. The obtained keypoint dataset is then used to train an "anticipation model" that predicts future keypoints (subgoals) for a low-level RL policy to execute. Experiments are conducted on a total of 5 tasks from Meta-World (3) and ManiSkill (2).

**Strengths:**

- I believe that this paper studies a relevant and timely problem (learning shaped reward functions from video demonstration data), and is likely to be of interest to the community. The problem is clearly motivated in the introduction, and shortcomings of existing work is described in the related work section. The paper is generally well written and easy to follow (exceptions to this are mentioned in the following sections of my review), and I believe that the technical contributions are original.
- The proposed method seems technically sound and appears to rely fairly little on engineering or manual labor (only exception to this is prompting for SAM as far as I can tell), which is definitely a plus. There is nothing too surprising about the approach in my view, but I consider that a strength! It seems likely that it would transfer to other settings than the specific task suites (Meta-World, ManiSkill) considered in the experiment section of the paper.
- I appreciate the inclusion of both online RL and offline RL settings as well as two different task suites (but I have some reservations as discussed in "weaknesses"), and I believe that the baselines considered are both appropriate and strong. The authors also used 5 random seeds in their experiments which makes me more confident in the results shown.

**Weaknesses:**

- Several aspects of the paper are in my opinion overclaimed: *(1)* not all robotic manipulation tasks can be represented as a set of keypoints and the approach thus has a narrower set of tasks for which it is appropriate compared to more general representation learning methods and human-engineered reward functions; for example, anything involving fluids, particles, or deformable objects would be difficult to represent using keypoints (pouring, scooping, wiping, etc.), as well as tasks for which the target is a certain *motion* rather than a state of the world in which everything is at rest. *(2)* the authors describe the method as if it is a multi-task reward (anticipation) model but it is unclear whether that is actually the case in their experiments, and either way I do not believe that the problem setting is sufficiently multitask if it involves only 3-5 tasks. *(3)* the authors claim that "Extensive experiments on complex, long-horizon manipulation tasks show that ReLAM significantly accelerates learning and achieves superior performance compared to state-of-the-art methods" (abstract) when in fact this only includes 5 simple manipulation tasks from Meta-World and ManiSkill both of which have short task horizons (push button, open drawer, etc.).
- As I allure to above, it is difficult to claim superior performance on long-horizon tasks when in fact only 5 tasks are considered and none of them are particularly long-horizon. I am willing to believe that the proposed method would work well for long-horizon problems, but there presently is little evidence of that. I would also strongly recommend the authors to consider more tasks both for reward learning and downstream policy learning benchmarking as that would make the results more convincing.
- It is unclear why the authors consider only online RL in Meta-World and only offline RL in ManiSkill when both task suites were designed for online RL originally. It would be more informative to compare online and offline performance on the same set of tasks.
- It appears that the authors do not use proprioceptive information in their approach nor any of the baselines. I can see how tracking the end-effector visually via keypoints will provide similar albeit more noisy information, but since the authors focus specifically on robotic manipulation for which proprioceptive information typically is available I would have expected that too. In particular, I believe that proprioceptive information would make grasping much easier.

Minor comments that need to be addressed but are insignificant wrt my overall assessment of the work:
- Axes labels and legends in figures 4,5,6 are way too small; the font size needs to be increased. I had to use 300% zoom for the text to be legible.
- The paper would benefit from a round of proof-reading, I found typos throughout the paper. For example, repeated use of "Orcale" instead of "Oracle" in e.g. L359 *"(4) Orcale replaces the generated image"* and Figure 4, as well as inconsistent use of terms (*e.g.* sometimes Meta-World, other times Metaworld).

**Questions:**

I would appreciate it if the authors could please address my comments in the "weaknesses" section above.

Additionally, I would like the authors to clarify whether the anticipation model used in the experiments is trained in a multi-task manner, I could not find any information about this except in L250: *"Note that ReLAM can be extended to multi-task scenarios by adding a task indication frame $I_{\text{task}}$ to the anticipation model’s input."* which leaves ambiguity.

---

### Official Review · Reviewer_wc8A · 2025-11-01

**Soundness:** 2
**Presentation:** 3
**Contribution:** 2
**Rating:** 4
**Confidence:** 4

**Summary:**

This paper tackles the difficulty of designing dense rewards for visual RL in robotic manipulation without precise state data. ReLAM framework learns an anticipation model from action-free videos to generate keypoint-based subgoals and derive continuous, geometry-aware rewards. This enables efficient hierarchical policy learning that aligns with task structure. Experiments on Meta-World and ManiSkill show that ReLAM greatly accelerates learning and outperforms prior video-based reward methods.

**Strengths:**

- Effectively leverages off-the-shelf vision models to design dense reward functions without requiring precise target object information.
- The anticipation model successfully predicts subgoals that closely resemble true subgoals in the task sequence.

**Weaknesses:**

- The number of experimental tasks is too small, and they are relatively simple. For instance, the three MetaWorld tasks used in the paper are known to be low-difficulty tasks within the MetaWorld benchmark [1]. Additional experiments on more diverse and challenging tasks are needed to validate the method’s effectiveness.
- The proposed anticipation model appears highly sensitive to visual variations, making it difficult to apply in real-world robotic settings, where factors like lighting and shading vary significantly more than in simulations. At minimum, experiments demonstrating visual robustness in simulated environments would strengthen the work.
- The use of hierarchical reinforcement learning (HRL) is typically motivated by long-horizon or sparse-reward tasks. However, most tasks in this paper are short-horizon (within ~100 timesteps) and relatively simple. Prior works [1,2] have shown that such tasks can often be solved with single-agent RL even without ground-truth rewards, suggesting that the proposed method may be overly complex for these cases. Demonstrating ReLAM’s effectiveness on more difficult, long-horizon tasks would make the contribution more compelling.
- Some tasks cannot be accurately represented by rewards based solely on distance to the target object. For example, in insertion tasks, even if the object moves correctly toward the target, successful completion depends on precise alignment (e.g., whether the object fits into a hole). It is unclear whether the current object segmentation and tracking pipeline can capture such fine-grained pixel-level differences.
- The assumption that motion between keyframes is linear overly constrains the diversity of possible robotic behaviors. For more complex tasks involving rotations or non-linear trajectories, ReLAM may be difficult to apply directly.
- The paper lacks citations and comparisons with relevant prior works [1,2], which would help contextualize its novelty and limitations.

**References**\
[1] Subtask-Aware Visual Reward Learning from Segmented Demonstrations, ICLR 2025.
[2] ReWiND: Language-Guided Rewards Teach Robot Policies without New Demonstrations, CoRL 2025.

**Questions:**

- Since the framework relies on off-the-shelf models, how long does reward inference take in practice? Additionally, what is the total training time for the agent when incorporating this process?
- I think the proposed tasks are too easy. Could you conduct some additional experiments with harder tasks (in the same Meta-World  or ManiSkill), and prove the efficacy of the proposed method?

---

### Official Review · Reviewer_Yy7r · 2025-11-02

**Soundness:** 3
**Presentation:** 3
**Contribution:** 2
**Rating:** 2
**Confidence:** 4

**Summary:**

This paper focuses on reward design in visual RL for robotic manipulation. It is motivated by the fact that precise positional information is often unavailable in real-world visual settings due to sensory and perceptual limitations. Therefore, this paper introduces ReLAM, a framework that automates reward design for visual robotic manipulation using only action-free video demonstrations, and it implicitly infers spatial distances through keypoints extracted from images.

ReLAM first trains an "anticipation model" to analyze videos and predict a sequence of keypoint-based subgoals, effectively creating a high-level plan for the task. A low-level robotic policy is then trained within a hierarchical RL framework, guided by a dense reward signal calculated from the distance to these keypoint subgoals.

This method simplifies the learning problem by abstracting visual scenes into keypoints, enabling faster training and superior performance on complex, long-horizon tasks compared to existing approaches.

**Strengths:**

- Automatic reward design is an important topic in RL for robot manipualtion. This framework only assumes unlabeled video demonstrations, eliminating the need for tedious manual reward engineering.

- ReLAM effectively decomposes long-horizon tasks by using a high-level anticipation model to generate a curriculum of subgoals and a low-level policy to execute them, improving learning efficiency and scalability.

- ReLAM outperforms several existing reward learning methods in complex robotic manipulation tasks, achieving higher success rates and faster learning.

- The approach is supported by a mathematical analysis that provides a provable sub-optimality bound, which addes theoretical rigor.

**Weaknesses:**

- The paper's primary motivation is to solve a real-world problem. **They claim the precise positional information used in traditional reward design is often unavailable in real-world visual settings due to sensory and perceptual limitations. However, this paper does not validate the proposed solution in the real world at all**. All the experiemnts are exclusively on simulated environments where the very ground-truth data it claims is unavailable, actually exists. I understand that setting up a real-world RL system is challenging. However, if the motivation of the whole paper is that the traditional rewards cannot be used in the real world, then it becomes necessary to show that the proposed reward learning approach actually works in the real world.

- The method for extracting keypoints relies on a brittle chain of models and heuristics: a text-prompted segmentation model (SAM), a tracking model, motion-based filtering, and Farthest Point Sampling. A failure at any stage (an incorrect segmentation, a lost track due to occlusion or motion blur, or a poorly chosen threshold) could cause the entire system to fail. This complexity undermines the claim of a robust, automatic framework.

- The framework is not fully automatic. The SAM model requires manual, task-specific text prompts (e.g., "green drawer," "blue cube"), which is a form of human supervision. Furthermore, the method relies on several "predefined" thresholds and parameters (e.g., for motion filtering, keyframe selection), which likely require careful tuning for each new task or environment, limiting its generalizability.

- The keyframe selection process is based on the assumption that robotic tasks can be decomposed into segments of "linear motion." This is a gross oversimplification. Many manipulation tasks involve complex, non-linear trajectories (e.g., turning a valve, wiping a curved surface), which this method would fail to model correctly.

**Questions:**

See weaknesses.

---

### Note · Authors · 2025-12-02

I have read and agree with the venue's withdrawal policy on behalf of myself and my co-authors.